# The Role of Innate and Adaptive Immune Cells in Skeletal Muscle Regeneration

**DOI:** 10.3390/ijms22063265

**Published:** 2021-03-23

**Authors:** Natalia Ziemkiewicz, Genevieve Hilliard, Nicholas A. Pullen, Koyal Garg

**Affiliations:** 1Department of Biomedical Engineering, Parks College of Engineering, Aviation, and Technology, Saint Louis University, 3507 Lindell Blvd, St. Louis, MO 63103, USA; natalia.ziemkiewicz@slu.edu; 2Department of Biology, Saint Louis University, St. Louis, MO 63103, USA; gen.hilliard@slu.edu; 3School of Biological Sciences, College of Natural and Health Sciences, University of Northern Colorado, Greeley, Colorado, CO 80639, USA; nicholas.pullen@unco.edu

**Keywords:** inflammation, myogenesis, mesenchymal stem cell, T cell, macrophage, immunomodulatory therapy

## Abstract

Skeletal muscle regeneration is highly dependent on the inflammatory response. A wide variety of innate and adaptive immune cells orchestrate the complex process of muscle repair. This review provides information about the various types of immune cells and biomolecules that have been shown to mediate muscle regeneration following injury and degenerative diseases. Recently developed cell and drug-based immunomodulatory strategies are highlighted. An improved understanding of the immune response to injured and diseased skeletal muscle will be essential for the development of therapeutic strategies.

## 1. Introduction

Skeletal muscle injury or disease can result in an inflammatory response in which phagocytes and lymphocytes rapidly invade the tissue to influence satellite cell activation, proliferation, and terminal differentiation (Figure 1) [1,2]. Immediately after injury, the muscle cells begin to undergo necrosis. A compromised sarcolemma causes the cellular contents to be released into the extracellular space. Infiltrating immune cells, such as neutrophils and macrophages, start clearing out the necrotic cell debris and also secrete various growth factors and cytokines to recruit more immune cells to the site of injury [3]. Neutrophils appear to play a transient role and are quickly replaced by macrophages, which persist for weeks to months (Figure 2). Macrophages exhibit diverse phenotypes and can promote either muscle injury or repair depending upon the severity of the injury and the timing of phenotypic (M1/M2) transition [4,5,6].

Macrophages are also believed to be responsible for recruiting T cells to the site of muscle injury [8]. The two main T cell types—helper (CD4) and cytotoxic (CD8) can be detected in the wound site as early as three days post-injury [9,10,11]. Similar to macrophages, T cells have been shown to persist at the site of injury for several weeks after trauma. The sustained presence of these cell types at the site of injury suggests that they are critically involved in the regenerative process.

In this work, we review various experimental studies that have attempted to distinguish between the features of the inflammatory response that promote muscle damage, fibrosis, or repair. Several cell types and biomolecules have emerged as targets for therapy design and the development of immunomodulatory strategies is an active area of research for skeletal muscle trauma. While it has been suggested that a heightened and prolonged inflammatory response is detrimental for muscle regeneration, it has also been shown that ablating inflammation is not an effective approach for muscle repair. The precise conditions in which inflammation results in a functionally beneficial regenerative response are unclear. Nevertheless, the activity of invading immune cell types and their actions on resident muscle stem cells largely dictate the regenerative outcome, and the purpose of this review is to highlight and describe the activity of key cellular players in the immune response to injured and diseased muscle tissue.

## 2. Innate Immune Cells Influence Muscle Regeneration

### 2.1. The Role of Neutrophils in Muscle Regeneration

Muscle damage is common and can result from sports-related injuries or diseases, such as muscular dystrophy. Mechanical changes that cause trauma or stress to muscle membranes, resulting from injury or overload, initiate a cascade of events that ultimately lead to granulocyte, primarily phagocytic neutrophil infiltration into the site of injury [12,13,14]. Perturbations of skeletal muscle propagates the release of pro-inflammatory cytokines such as tumor necrosis factor α (TNF-α) and interleukin (IL)-1β [12]. These cytokines act on the endothelium of adjacent vasculature causing the release of IL-6 and IL-8 (also known as CXCL8 or neutrophil chemotactic factor) [3,12]. These cytokines, especially IL-8, serve as chemoattractants for neutrophils and cause their migration from circulation into areas of damage [3]. Neutrophils can appear as early as one hour after muscle injury [15,16] and persist for up to a week [17,18].

Their primary role in injured tissue is to remove cellular debris through phagocytosis and recruit macrophages to the wound environment [12,14]. Once activated, neutrophils generate a robust inflammatory response through the release of cytokines (e.g., TNF-α, IL-1, and more IL-8) [19], proteases, and reactive oxygen species (ROS; ex: O₂ˉ, H₂O₂, OHˉ, HClO) [20]. The removal of cellular and fiber debris facilitates muscle regeneration and connective tissue deposition [12]. Additionally, neutrophils can secrete chemoattractants to recruit macrophages to the site of damage. Macrophage infiltration can influence various steps in the process of muscle regeneration, which are described in detail below. Interestingly, macrophage mediated clearance of apoptotic neutrophils stimulates the secretion of TGF-β1 and IL-10, and polarizes the macrophages to an anti-inflammatory phenotype [21]. Besides muscle fiber regeneration, neutrophils can also impact nerve healing and removal of neutrophils may be detrimental to reinnervation [22]. The role of neutrophils is not fully understood, but there is some evidence from a variety of systems to suggest that they can also promote tissue regeneration through regulating angiogenesis and lymphangiogenesis [23,24].

Further evidence for the critical role of neutrophils in muscle regeneration is provided by studies where neutrophils were depleted. For instance, mice injected with a snake venom toxin had significant necrosis and an impaired regenerative response in the absence of neutrophils [25]. Other studies have suggested that neutrophils do not significantly impact muscle recovery. In one research study, animals were subjected to ten days of hind limb unloading, followed by depletion of neutrophils (using the Ly6G/Ly6C (Gr-1) antibody) and then subsequent reloading [26]. The results showed that neutrophil depletion had no impact on the loss of force or recovery of the atrophied fibers [26]. However, it should be noted that incubating soleus muscle after unloading/loading with LPS, a neutrophil activator, led to a 20% greater deficit in force, as well as increased muscle damage, when compared to muscle with depleted neutrophils. The potential for neutrophils to worsen muscle damage and exacerbate injury has also been studied extensively. For example, neutrophil-derived mediators can cause myofiber lesions, membrane lysis, and the oxidative degradation of lipids [2,12,27,28]. While muscle cells will release protective factors and antioxidants such as superoxide dismutase 2 (SOD-2), glutathione peroxidase (GPX), catalase (CAT), and thioredoxin (TRX) which will shield them from the detrimental effects of neutrophil oxidants, overproduction of cytolytic and cytotoxic compounds by neutrophils can further injure the already damaged surrounding tissue [2,29,30].

Substantial research focused on the impact of neutrophils in muscle damage began in the 1980s, and strong evidence suggests that they hinder rather than help muscle regeneration. One of the earliest studies conducted on the topic demonstrates that blocking neutrophil adhesion activity with the anti-CD11b antibody M1/70 reduces oxidant production and myofiber damage after stretch injury at early time points [31]. Blocking CD11b or CD11a has had beneficial effects in a traumatic skeletal injury model induced by a needle puncture as well, with a general reduction in myofiber damage [32]. Other methods of inhibiting neutrophil activity after exercise also mitigate overall muscle cell damage. For example, inhibiting neutrophil activity by mutating the *gp91-phox* gene, which ultimately leads to superoxide production, protects fibers from damage after hind limb unloading, with a significant reduction in membrane lysis [33]. In a similar study of loading and unloading exercise, mice deficient in neutrophil-derived myeloperoxidase enzyme (MPO) enzyme had significantly less myofiber lysis [34].

Further support for the detrimental role of neutrophils came from a study that reportedly depleted them through with a Ly6G antibody to block activity after exhaustive exercise, and this resulted in reduced myofiber lesions [35]. These observations were correlated with a reduction in TNF-α, IL-6, and macrophage infiltration providing evidence that these are some of the mediators by which neutrophils can cause tissue damage [35]. Neutrophils also mediate destruction through the β2 integrin receptor CD18 and inhibiting this receptor led to a decrease in force deficit and overall oxidative damage with a concomitant increase in the proportion and cross-sectional area of regenerating fibers [15].

The role of neutrophils in exacerbating ischemia reperfusion (I/R) injuries has been well documented [36]. At the site of I/R injury, neutrophils release ROS, proteinases, proinflammatory cytokines and chemokines [2]. Together, these mediators can increase the immune response and lead to tissue necrosis. Once again, the depletion of neutrophils reduced the extent of injury [37,38] and functional deficits [39] following I/R injury.

Besides injuries, neutrophil-mediated muscle damage is also heavily implicated in myopathies, such as muscular dystrophy that show chronic and persistent inflammation [29]. Neutrophil elastase and ROS were found elevated in an animal model of muscular dystrophy (*mdx*) and contribute to a reduction in myoblast survival and differentiation [40]. Depleting neutrophils in young *mdx* mice reduced muscle breakdown and blocking TNF-α activity with etanercept decreased exercise-induced muscle damage in adult *mdx* mice [41]. Utilizing other TNF-α blockers, such as infliximab, also proved beneficial and reduced muscle damage, as evidenced by reduced inflammation and necrosis, along with increased myotube formation [42]. Collectively, these results suggest that either removing neutrophils or blocking mediators of inflammatory activity, such as TNF-α, has beneficial effects on muscle regeneration and supports the hypothesis that neutrophils harm muscle.

Neutrophils have also been implicated in the age-associated decline in muscle regeneration. Neutrophils and macrophages often work in concert to promote the immune response following injury [43]. Similar correlation has been found in aged animals following contusion injury, where an increase in macrophage and neutrophil populations was found [44,45]. However, some studies have highlighted a decline in cellular activity with aging. For instance, a lower phagocytic capacity of macrophages was implicated in the slowing down of the regenerative process [46].

Based on the review of published literature, it appears that neutrophils play dual roles in the muscle regeneration process where they can promote muscle damage and contribute to muscle repair through different mechanisms.

### 2.2. Macrophages

Macrophages are highly versatile innate effector cells that play a critical role in the mounting and resolution of inflammatory responses [47,48]. Monocytes, originating in the bone marrow, differentiate into macrophages once they reach the tissue and are influenced by the inflammatory milieu and pathogen-associated pattern recognition receptors (PRR) [49,50]. Both tissue-resident and recruited macrophages play important roles in skeletal muscle tissue repair following injury [1,51,52]. While resident macrophages act as initial sensors of pathological events, recruited macrophages amplify the inflammatory response and form a link between innate and adaptive immunological responses [53]. Recent studies further highlight that resident and recruited macrophages are developmentally and functionally distinct [54,55]. Macrophages have several functions in muscle repair and remodeling, including phagocytosis, enzyme secretion, cytokine and growth factor production, antigen presentation, and immune cell recruitment.

Macrophages exhibit complex and hybrid phenotypes as a result of the wide range of activation states they experience [56]. According to a nomenclature introduced in 2000 [57], macrophage activation states may include “classically activated” (M1) macrophages that are pro-inflammatory and “alternatively activated” (M2) macrophages that are anti-inflammatory [52]. Under in vitro conditions, the activation state of macrophages was observed to be analogous to the helper T cell type 1 (T_h_1) and type 2 (T_h_2) observed for T lymphocytes (T cells); however, in vivo macrophages demonstrate a wide variety of phenotypes based on the complexity of signals present in the tissue [52,56,58]. In response to LPS treatment or interferon γ (IFN-γ), macrophages become classically activated. M1 macrophages are characterized by the expression of inducible nitrogen oxide synthase (iNOS), pro-inflammatory cytokines (e.g., TNF-α, IL-1β, IL-6, IL-12), CD68, and Toll-like receptor (TLR) ligands. In response to IL-4 and IL-13 acting on common receptor chain, IL-4Rα, macrophages become alternatively activated [59]. M2 macrophages are characterized by the expression of arginase-1 (ARG1), scavenger receptors (CD163), mannose receptors (CD206), and/or anti-inflammatory cytokines (e.g., IL-10) [5,47,48,60,61,62]. Mantovani et al. further subdivided M2 macrophages into M2a, M2b, M2c, and M2d subtypes based on their function and how they are induced by different stimulatory factors. The M2a phenotype is obtained when macrophages are stimulated with IL-4 and IL-13 and are also named wound-healing macrophages for their anti-inflammatory function. M2b macrophages, named regulatory macrophages, play a role in immunoregulation by monitoring the extent of the immune response and the inflammatory reaction. M2b macrophages are induced when exposed to immune complexes (IC) and TLR agonists or IL-1 receptor (IL-1R) agonists. Stimulating macrophages with IL-10 and glucocorticoids induces the M2c phenotype aiding in immunosuppression, phagocytosis, and tissue remodeling [4,62,63,64,65,66]. M2d macrophages, a novel subset of macrophages considered tumor-associated macrophages (TAMs) and are induced by the co-stimulation of TLR ligands and A2 adenosine receptor (A2R) agonists or IL-6. TAMs have been found to contribute to angiogenesis and cancer metastasis while also playing a critical role in the inflammatory response of neoplastic tissue [63,67,68]. While these macrophage phenotypes have been extensively characterized in vitro, their role in regulating tissue regeneration in vivo is unclear. It should also be noted that the M1 and M2 polarization states are hypothetical ends of a spectrum of activation states in which macrophages can exist. The in vivo polarization state of macrophages can be highly complex and heterogeneous with a high degree of plasticity.

Skeletal muscle has remarkable regenerative capabilities following injury. After an injury, a plethora of events occurs, including the activation of satellite cells (MuSC) (Pax7^+^) and the infiltration of immune cells, such as neutrophils and macrophages [47,69]. Several studies have demonstrated the importance of macrophages in the regeneration of skeletal muscle using various injury models (e.g., hind limb ischemia, freeze-injury, unloading/loading sequences, myotoxic agent injection) [1,48,51,59,69,70,71,72]. After the initial neutrophil influx at the site of injury, macrophages become the dominant inflammatory cell type with peak levels at day 3 post injury [1,47,71].

The recovery of skeletal muscle requires macrophages to phagocytize and remove cellular debris from necrotic muscle tissue, while also producing a vast array of chemokines and cytokines for recruitment of stem cells and coordination of the repair process. The activation state of macrophages influences the myogenic process by acting on myogenic precursor cells. In co-cultures, pro-inflammatory M1 macrophages were more influential in enhancing myogenic precursor cells proliferation, whereas anti-inflammatory M2 macrophages enhanced differentiation, indicated by myogenin expression and myotube formation [59]. Another study demonstrated the supportive role pro-inflammatory M1 macrophages can play in vivo when injected together with myoblasts. The study found that the co-delivery of these cells increased the efficiency of myoblast engraftment and played a role in muscle regeneration by increasing migration and proliferation events but delaying differentiation [73].

Besides paracrine influence on cellular function, macrophages can also have direct cell-to-cell interactions. One study demonstrated that myogenic precursor cells were rescued from apoptosis through direct contact with macrophages in vitro and in vivo [74] via adhesion molecules, such as intercellular adhesion molecule-1 (ICAM-1; CD54) and platelet endothelial cell adhesion molecule-1 (PECAM-1; CD31) [75].

### 2.3. Role of Macrophages in Muscle Injury

Muscle repair and regeneration following an acute muscle injury rely on an inflammatory response which activates a series of interactions between immune cells and muscle cells. The initial response mimics a T_h_1-like inflammatory response, with an influx of neutrophils, followed by CD68^+^ macrophages [1]. The presence of IFN-γ and TNF-α promotes classical activation of Ly6C^+^ monocytes into the M1 phenotype macrophages [1,60]. Monocytes are recruited to the site of injury by cells of both the innate and adaptive immune system, such as neutrophils and CD8^+^ T cells, and are classified by the expression level of Ly6C [76]. The interaction of CCR2/CCL2 and CX3CR1/CX3CL1 aids in the recruitment and differentiation of monocytes into macrophages [59,77,78,79].

Cytotoxic (CD8^+^) T cells facilitate the expression of CCL2 by resident macrophages in injured muscle, aiding in the recruitment of inflammatory monocytes [80]. Murine models deficient in CCR2 experienced decreased monocyte infiltration impairing angiogenesis and muscle regeneration whilst promoting adipocyte accumulation following a cardiotoxin-induced injury [81]. Another study showcased that the presence of CCR2 is essential to muscle regeneration, independent of TLR signaling, age, and sex in murine models as the decreased infiltration of monocytes/macrophages promoted a pro-inflammatory microenvironment [82]. Another study using CD8 knockout mice showed decreased expression of CCL2 resulting in impaired muscle regeneration and increased fibrosis [80]. Studies in mice lacking CCR2 or CCL2 have shown that monocyte/macrophage infiltration is impaired, altering the regenerative process directing myogenic precursor cells to promote an adipogenic phenotype [83,84,85,86,87]. The role of CX3CR1/CX3CL1 in skeletal muscle regeneration is still not well studied; however, one study found that a CX3CR1 deficiency rescues impaired muscle regeneration in CCL2 deficient mice following a notexin-induced muscle injury [77]. Another study also found that a CX3CR1 deficiency affected macrophage phagocytic functions, decreased insulin-like growth factor 1 (IGF-1) and IL-6 expression by macrophages, and may have impaired myogenic precursor cell differentiation [70].

The differentiation of monocytes to macrophages is critical to an appropriate healing response. In one study, the monocyte/macrophage population was depleted at various stages before and after cardiotoxin-induced muscle injury using the CD11b-diphtheria toxin receptor (DTR) transgenic mice. The results suggested that early ablations (days 0–2) resulted in greater impairment in regeneration, whereas later ablation (day 4) had a minimal effect [48]. Early ablation resulted in deficient muscle regeneration due to a decreased number of macrophages present at days 2 to 4, highlighting a role of pro-inflammatory macrophages in tissue repair [48]. M1 macrophages are typically associated with exacerbating tissue damage, with the potential of provoking a fibrotic healing response [88,89]. However, a study found that treatment of lacerated mouse muscles with exogenous M1 macrophages reduced fibrosis and enhanced muscle regeneration, highlighting a beneficial role for M1 macrophages in tissue repair [90]. In support, another study also showed that in vitro polarized M1 (LPS/IFN-γ) macrophages may positively influence muscle regeneration when injected into the gastrocnemius muscles after a tourniquet-induced ischemia-reperfusion (TK-I/R) injury. It was shown that the early delivery of M1 macrophages improved functional recovery, accelerated myofiber repair, decreased fibrotic tissue deposition, and increased whole muscle expression of IGF-1 following TK-I/R injury [91].

As mentioned earlier, macrophage phenotypes (M1/M2) play distinct but complementary roles in skeletal muscle healing. In vitro and in vivo, M1 macrophages promote muscle damage through the release of ROS and production of nitric oxide (NO) [2,12,92]. The concentration of NO present at the site of injury plays a role in its function; at high concentrations, apoptosis may be induced, while at low concentrations it can help protect cells against oxidative damage [93,94,95]. One study showed that NO activation was critical in the early phases of skeletal muscle repair in a murine muscle crush injury, as it increased MuSC proliferation and quantity [96,97]. The study demonstrated that the released NO may play a role as a regulator between skeletal muscle regeneration and fibrosis [97].

Once pro-inflammatory (or M1-like) macrophages reach their peak concentration, the pro-inflammatory microenvironment begins to convert into an anti-inflammatory microenvironment with a higher concentration of M2-like macrophages. Recently, studies have found that Forkhead box p3 (Foxp3^+^) CD4^+^ regulatory T cells (T_reg_) play a role in inducing the M2 phenotype and aid in the shift from M1 macrophages to M2 macrophages by producing IL-10 [98,99]. The study also showed that Tregs are recruited to the site of injury by CCL2, similar to monocytes [99]. Studies show that the switch from M1 macrophages to M2 macrophages is a crucial step in skeletal muscle regeneration. The role of IL-10 in mediating the transition between M1 to M2 macrophages was investigated in mice experiencing muscle reloading after the induction of disuse atrophy due to hindlimb unloading. IL-10 ablation significantly reduced the expression of M2 macrophage markers (i.e., CD163 and Arg 1) following 4 days of reloading, indicating a key role in inducing M2 polarization. The study also found that, without the induction of M2 macrophages, muscle regeneration was reduced, and muscle fiber growth was significantly slowed [100]. The beneficial effect of M2 macrophages has also been demonstrated following endurance exercise training in humans [101]. A significant increase in M2 macrophage quantity after 12 weeks of endurance exercise training was associated with myofiber hypertrophy and satellite cell accumulation.

Therefore, the transition to an M2 macrophage phenotype is believed to be essential for effective and complete tissue regeneration. A great deal of effort is being made to tissue engineer therapies that will rapidly elicit an M2 macrophage phenotype upon transplantation. For instance, scaffolds that release IL-4 [102,103], IL-10 [104,105] or present large enough pore-sizes [106], are being designed with the goal to promote an M2 macrophage phenotype in vivo.

At least one study has highlighted the importance of timing at which the polarization of M1 and M2 macrophages takes place. It was shown that introducing M1 macrophages in a temporally coordinated manner can improve angiogenesis and skeletal muscle regeneration following hind limb ischemia [107]. The study further concluded that the persistent activation of M1 macrophages may exacerbate muscle tissue damage; however, the premature introduction of M2 macrophages in the acute inflammatory phase can interfere with muscle regeneration, impeding muscle repair while promoting fibrosis [107].

Regeneration of skeletal muscle following chronic muscle injuries varies greatly from the healing response of an acute muscle injury. The inflammatory response following a chronic muscle injury differs from the cellular response observed in muscle injuries such as crush, freeze, or toxin-induced injuries. Genetic dystrophies are associated with an inflammatory component, evoked by DAMP release due to muscle fiber damage sustained during muscle contraction, and this ultimately results in fiber degeneration and loss of motor unit function [11]. In one study, IL-10 expression was ablated in *mdx* mice resulting in increased muscle damage and decreased strength [108]. The results of this study suggest that IL-10 plays a role in deactivating M1 macrophages, attenuating the dystrophic pathology at early, acute stages of the disease [108]. One study demonstrated that the depletion of macrophages in an *mdx* murine model at the early, acute peak of muscle pathology resulted in large reductions of lesions in the sarcolemma of muscle fibers [109]. Taken together, these studies demonstrate that M1 macrophages play a detrimental role in muscular dystrophy due to their highly cytolytic nature, but a reduction in muscle damage may be observed through M2 phenotype switch [110,111,112].

A defective MPC response, infiltration of macrophages, and heightened proliferation of matrix-producing cells are characteristics of chronic inflammation. Similar to muscular dystrophy, VML also exhibits chronic inflammation with repeated degeneration–regeneration cycles [113]. Studies working with VML defects in both rats and pigs found that macrophages persisted in the defects for weeks to months, unlike the immune response associated with an acute injury [113,114]. The role of extracellular matrix (ECM) scaffolds in modulating macrophage phenotypes (M1/M2) has been extensively studied in the context of VML. While some studies have shown positive regenerative outcomes with the induction of an M2 macrophage phenotype, others have highlighted that synergistic roles played by both M1 and M2 phenotypes that are critical for regeneration [115,116]. Following autograft treatment of VML injury, mixed upregulation of pro- and anti-inflammatory macrophage phenotypes is observed for several weeks post-injury. This mixed macrophage phenotype was associated with improved regenerative outcomes and muscle function.

### 2.4. Role of Macrophages in Aged Muscle

Skeletal muscle undergoes several cellular and metabolic changes with aging, but the underlying mechanisms are poorly characterized. Skeletal muscle aging is strongly influenced by the imbalance of the damage and regeneration processes, at the molecular and cellular levels [45,117,118,119]. The immune system is known to experience changes in function with aging, and one study showed that factors released by young but not old bone marrow cells supported myoblast proliferation and differentiation in vitro [120].

The predominant phenotypes of cells could also change with aging. In a study comparing subpopulations of macrophages in young (average 31.9 years) and elderly individuals (average 71.4 years), the results showed that the quantity of pro-inflammatory M1 macrophages (CD11b^+^) was lower in elderly muscle at baseline and also in response to resistance exercise when compared to younger muscle, while M2 macrophages (CD163^+^) were more prevalent [121]. M2 macrophages are the major type of macrophage in healthy skeletal muscle, while M1 macrophages are typically lower in abundance. Due to the low-grade chronic state of inflammation observed in aged muscle, one would expect to see an increase in M1 macrophages and decrease in M2 macrophages; however, a study found that M2 macrophages (CD206^+^) increased with age while M1 macrophages (CD80^+^) declined with age [122]. Another study comparing young (21–33 years) and elderly individuals (70–81 years) showed similar results. Higher CD206 gene expression was observed in the muscles of elderly individuals, while no difference in total macrophage content (CD68^+^) was detected. CD206, a M2 macrophage marker, is suggestive of an increase in the proportion of anti-inflammatory macrophages in senescent muscle [123,124]. One study showed an increase in IL-4 driven M2a macrophages (CD163^+^) in aged muscles, which were implicated in muscle fibrosis. Loss of NO production in aged muscle enabled the rise of M2a macrophages, resembling the mechanism driving fibrosis in *mdx* mice [109,125,126]. Collectively, these studies highlight a shift in M1/M2 phenotype balance with age, and a potential detrimental role of increased M2 macrophage accumulation.

Interestingly, it has also been shown that ECM scaffolds derived from animals show different remodeling outcomes and macrophage polarization depending on the age of the source animal. It was found that ECM scaffolds from younger pigs (3–12 weeks old) stimulated a predominantly M2 macrophage response compared to older pigs (26–52 weeks old). An M2 macrophage response was associated with greater myogenesis and lower fibrosis [127]. Therefore, age as a biological variable should be accounted for when studying macrophage polarization.

## 3. Adaptive Immune Cells Mediate Muscle Regeneration

### 3.1. Helper and Cytotoxic T Cells

T and B lymphocytes are two of the main cell families that comprise the adaptive immune system, which generates hyper-specific memory responses to immunological threats. This review focuses on the T cells, which consist of CD4^+^ (helper) and CD8^+^ (cytotoxic) groups. The CD8^+^ group is important for clearance of intracellular pathogens (e.g., viruses, mycobacteria, etc.) and emergent neoplasms by direct cell killing. The CD4^+^ population is the most abundant, well-characterized, and impactful group on overall immune function as their name, helper, indicates their coordination of all immune responses. This help to all the other cells of the immune system is proffered by the cytokine milieu produced by different effector phenotypes of CD4^+^ T cells, which includes T_h_1, T_h_2, T_h_17, T_fh_, and T_reg_ subsets. The help these cells offer is tuned to the type of threat or stage/progress of immune response. These phenotypes are well-characterized and detailed elsewhere [128]. Many of the cytokine signals that potentiate or come from CD8^+^, T_h_1 and T_h_17 cells polarize macrophages toward a pro-inflammatory M1 phenotype, while T_h_2 and T_reg_-inducing cytokines promote an anti-inflammatory M2 phenotype. The timing and sources of cytokines eliciting macrophage and T_reg_ responses may be tissue-dependent and is an important line of inquiry requiring further research [129,130,131].

CD8^+^ T cells develop into cytotoxic T cells which can directly destroy infected cells by inducing apoptosis via release of contents from cytolytic granules and/or interaction with death receptor Fas on target cells by Fas Ligand on the CD8^+^ [128]. The T cell granules contain perforins, cytolysins, proteases, and granulolysins that are all designed to degrade a cell [128]. The main focus of the review is pro-inflammatory CD8^+^ and CD4^+^ (T_h_1, T_h_2, and T_h_17) and immunosuppressive regulatory CD4^+^ (T_reg_) cells as they relate to muscle physiology; however, B cells and Natural Killer cells have been found in myopathies and post-exercise, respectively [132,133,134].

### 3.2. T Cell Response to Muscle Injury

Growing evidence suggests that T cells can mediate tissue repair by influencing cell types such as macrophages, stem cells, and myoblasts which act in concert to promote muscle fiber regeneration. They also secrete a wide variety of factors that can skew the healing environment towards either a pro-inflammatory or pro-regenerative response.

In a cardiotoxin model of muscle injury, T cells facilitated regeneration and their absence dramatically hindered recovery. Mice lacking CD8^+^ T cells (CD8^-/-^) after cardiotoxin injury showed markedly less regeneration as evidenced by smaller cross-sectional area (CSA) and heightened matrix deposition [80]. When T cells were then transplanted into CD8^-/-^ animals, regeneration was enhanced [80]. It is likely that the improvement is due to the presence of Gr1^High^ macrophages, which are recruited to the muscle by monocyte chemoattractant protein 1 (MCP-1), released by CD8^+^ T cells [80]. As Gr1^High^ macrophages promote MuSC proliferation, the stem cell pool decreases in the absence of T cells leading to diminished regeneration [80].

In another model of cardiotoxin injury, Rag1^-/-^ mice, which lack T and B-cells, were subjected to injury and displayed significantly delayed regeneration marked by a reduction in fiber organization, size, and myofibers with centrally located nuclei (CLN) [135]. These detrimental effects were recovered after transplanting activated CD3^+^ T and B-cells into injured Rag1^-/-^ mice [135]. In addition to mediating macrophage activity, the negative side effects of removing T cells may be partly due to reduced stem cell and myogenic activity in their absence. Activated T cells or T cell conditioned media cultured with muscle stem cells caused an increase in the proliferation of stem cells that were Pax7^high^ and MyoD^low^ [135]. Likewise, conditioned media collected from IL-2 and CD3 activated lymphocytes was cultured with myoblasts and they exhibited reduced differentiation, with downregulation of myoblast determination protein 1 (MyoD) and myogenin [136].

Similar effects have been seen in multi-muscle injury models, in which the extensor digitorum longus (EDL) and tibialis anterior (TA) are injured with a cardiotoxin injection. In these injuries, blocking IFN-γ receptor signaling with an antibody negatively impacted muscle regeneration [137]. CD4^+^ T cells, natural killer cells, macrophages, and myoblasts express IFN-γ during muscle injury, and blocking the IFN-γ receptors on these cell types resulted in decreased fiber regeneration, cellular proliferation, and MyoD^+^ cells [137]. In vitro, blocking the IFN-γ receptor on C2C12 cells caused a decrease in proliferation and myoblast fusion providing further evidence of the positive impact that T cells contribute to muscle regeneration through the release of their cytokines [137].

In composite injuries consisting of VML defects and open tibia fractures, the extent of damage can prolong T cell presence past the 3-day mark where they are initially found significantly elevated in VML injuries [138]. VML injury impaired bone regeneration and maintained elevated levels of both CD4^+^ and CD8^+^ T cells at 3, 14, and 28 days post-composite trauma [9]. Minced muscle graft (MMG) treatment attenuated the inflammatory response generated by composite injuries and reduced CD4^+^ T cell but not CD8^+^ T cell infiltration into the defect site 3 days post-injury [138]. MMG treatment had an overall depressive effect on the inflammatory response and encouraged regeneration through factor release (e.g., MCP-1, IL-10, and IGF-1) and satellite cell delivery [138].

It has also been shown that in treating muscle with bioengineered scaffolds, T cells are necessary for proper healing. In a traumatic muscle injury model, in which a portion of the quadriceps was excised and subsequently treated with either a bone or cardiac derived ECM scaffold, T cells were significantly elevated in response to the scaffold [139]. The ECM scaffolds reportedly modified the T cell composition towards a pro-regenerative CD4^+^ T_h_2 cell response [139]. T_h_2 cytokine IL-4 was also present in scaffold-treated wounds, while T_h_1 cytokine genes *Ifng, Fasl, Cd28, and Tbx21* were downregulated [139]. Associated with the T_h_2 response was an inhibition of M1 macrophages and activation of M2 macrophages, which are essential for muscle regeneration [139]. CD4^+^ T cells were also found to be important in the healing of ischemia-induced muscle injuries. In a model of ischemia, the gastrocnemius, plantaris, and soleus muscles were treated with alginate gels containing conditioned media from different subsets of CD4^+^ T cells (T_h_1, T_h_2, T_h_17, and T_reg_) [140]. By mixing cytokines from the different subsets of cells, it was found that conditioned media from differing combinations of CD4^+^ T cell subsets could regulate angiogenesis and myogenesis [140]. For instance, T_h_2 and T_h_17 cytokines augmented angiogenesis while T_h_1 cytokines induced vascular regression. A combination of T_h_1, T_h_2, and T_h_17 cytokines supported MPC proliferation but not differentiation. T_reg_ cytokines had little to no effect on angiogenesis or myogenesis in vitro or in vivo in this model.

In support, other studies have also shown that T cells may play a supportive role in muscle regeneration through interaction with MPCs. Activated T cells release several growth factors that can influence MuSC activity, including but not limited to fibroblast growth factor-2 (FGF-2), IFN-γ, TGF-β, TNF-α, IL-1, IL-4, and IL-13 [135,141,142]. One study identified that the combination of IL-1, IL-13, TNF-α, and IFN-γ pro-inflammatory cytokines, played a role in maintaining MPC potency and stimulating MPCs to proliferate in vivo [135].

An over-abundance of T cell infiltrate can be detrimental to tissue regeneration. The effects of this were illustrated in a study that induced TA injury via a cardiotoxin in Cbl-b (ubiquitin ligase) deficient mice, which increased the activation and infiltration of macrophages [8]. Cbl-b^-/-^ mice displayed poor healing, as indicated by increased fibrosis and inflammation [8]. As the ubiquitin ligase also helps regulate CD8^+^ T cells, the presence of the T cells was increased in the injured muscle [8]. The authors also identified that RANTES (CCL5), a chemokine secreted by macrophages, delayed T cell clearance and allowed them to persist for up to two weeks after injury [8]. Blocking RANTES activity with a neutralizing antibody rescued regeneration, leading to reduced CD8^+^ T cell infiltrate and fibroblast aggregation [8].

It should be noted that most of the models presented here are acute injuries, and in more chronic injuries, as seen with composite bone and muscle damage [10], T cell presence can be prolonged and potentially contribute to further damage. Therefore, when selecting therapies for muscle, T cells may present a good target to reduce inflammation and aberrant regeneration in chronic injuries.

### 3.3. T Cell Response to Exercise

Exercise-induced muscle damage can increase T cell presence in skeletal muscle. However, the extent of T cell infiltration and activity appears to be highly dependent on the type, intensity, and duration of exercise. For instance, strenuous eccentric cycling for 30 min did not increase T or B cell detection in skeletal muscle [143]. In another study, human subjects undergoing 45 min uphill or downhill running exercises did not experience significant differences in inflammation compared to non-exercised controls. Only the downhill running group reported delayed-onset muscle soreness (DOMS) with elevated T cells, neutrophils, and macrophages compared to the non-DOMS experiencing group. However, neither group was significantly different from the control group [144]. In contrast, other studies have reported heightened T cell presence following exercise. For example, T cell levels were found elevated in experienced athletes participating in an ultra-endurance exercise bout for 24 h. The numbers of total T cells, CD8^+^ cells, and macrophages (CD68^+^) were found to be 2–3 fold higher after 28 h of exercise [145]. Along with an increase in T cell populations, major histocompatibility class 1 (MHC I) expression was increased in muscle fibers. As healthy skeletal muscle normally shows very low MHC class I expression, it is possible that damaged muscle increases MHC class I expression to communicate with CD8^+^ T cells [145]. In another study investigating lengthening contractions, human participants underwent two sessions of exercise. It was found that CD8^+^ T cells and CD68^+^ macrophages infiltrated skeletal muscle only after the second session, with a concomitant increase in cytokines such as MCP-1 and IFN-γ-induced protein 10 (IP-10) [146]. The increased presence of CD8^+^ T cells was implicated in muscle adaptation to repeated eccentric contractions [146,147]. The study used DOMS as an indirect indicator of muscle damage, which was found significantly reduced after the second bout. As CD8^+^ T cells were increased significantly only after a second bout of exercise when evidence of muscle damage was reduced, it was suggested that CD8^+^ T cells do not exacerbate injury but facilitate repair [147,148]. Interestingly, this study reported no significant increase in MHC I following muscle damage.

Similar results were reported in another study with elite athletes that performed two bouts of high-intensity endurance exercise [149]. Higher concentrations of neutrophils, lymphocytes, T cells (CD4^+^ and CD8^+^) as well as natural killer (NK) cells (CD56^+^) were reported after the second bout of exercise. The study also reported a reduction in the percentage of CD56^+^CD69^+^ cells suggesting reduced state of NK cell activation and cytolytic activity.

In support, other studies have also indicated that NK cell cytolytic activity decreases after prolonged, intense, and stressful exercise [150]. These effects are possibly due to a temporary depression of the immune system post-exercise due to the presence of IL-6 in combination with anti-inflammatory cytokines such as IL-10, and interleukin 1 receptor antagonist (IL-1ra), which lead to decreased T_h_1 mediated responses [150]. However, there is controversy over the conclusions from these results and an extensive review challenged the role of exercise in immunosuppression [151]. Furthermore, in mouse models of skin and lung cancer there was an enhanced NK cell response with exercise. While this model has its challenges because the mice could wheel run ad libitum, there was a beneficial effect of exercise, and it was dependent on IL-6 (presumably muscle-derived) and epinephrine [152]. A similar study using a breast cancer mouse model supports this immune-potentiating effect by showing that exercise reduced myeloid-derived suppressor development [153].

Taken together, these studies highlight the need to investigate the role of T cells and their subsets not only in muscle repair following contraction induced injury but also in mediating the repeated bout effect. As described in the previous section, T cell derived factors can stimulate muscle stem cell proliferation. Therefore, future studies should explore the regulation of muscle stem cell activity via specific T cell subsets as a potential mechanism for muscle repair and adaptation following exercise.

### 3.4. Cytotoxic T Cells Mediate Continued Pathology Following Disease and Aging

Research confirming the deleterious role of T cells in dystrophy models is extensive and studies focused on depleting T cells demonstrate that their removal can dramatically improve muscle fiber regeneration. T cells account for roughly 3% of infiltrating cells in *mdx* models of dystrophy, with CD4^+^ and CD8^+^ T cells making up almost half of that proportion [154,155]. CD4^+^/CD8^+^ T cells have been found in dystrophic muscle at disease onset, with activated (CD44^high^) T cells present in both muscle and blood [154,156]. Muscle fibers invaded by CD8^+^ T cells typically express MHC-I, although the expression does not seem to be necessary for T cell-mediated cytotoxicity [157]. CD4^+^ T cells tend to dominate in dysferlinopathy, which is accompanied by the presence of membrane attack complex on the sarcolemma of fibers [158].

In an A/J limb/girdle dysferlin dystrophy- severe combined immunodeficiency (SCID) mouse model, the absence of T and B lymphocytes resulted in significantly improved muscle regeneration along with a concomitant increase in force production [159]. In the *mdx/scid* mouse model, depleting T and B lymphocytes led to a significant decrease in the expression of transforming growth factor β (TGF-β), a growth factor heavily implicated in fibrosis, accompanied by a decrease in fibrosis [160]. While these effects did not lead to any differences in myofibers with CLN, necrosis, degeneration, or muscle force compared to *mdx* controls, the research highlights the impact of T cells on fibrosis development [160].

Research targeting specific T cell populations for depletion has shown that CD8^+^ T cells are the main culprits of continued muscle degeneration. Depleting CD8^+^ T cells in *mdx* mice (by crossing them with perforin knockout mice) resulted in a marked reduction in apoptotic myonuclei and necrosis [154]. Interestingly, *mdx* mice that also lacked perforin experienced minimal apoptosis and macrophage invasion into the connective tissue, providing support that T cells cause damage through perforin [154]. Depleting CD8^+^ T cells can also lead to a reduction in muscle pathology as measured by areas of inflammation as well as necrotic fibers [156]. CD4^+^ T cells are also found in *mdx* muscle, and antibody depletion resulted in decreased muscle pathology similar to that seen after depleting CD8^+^ T cells [156]. In another study, CD45RC^high^ T cell depletion through a monoclonal antibody resulted in increased strength as measured through a grip test [161].

PKC theta, a regulator of T cell activation and proliferation, was knocked out in *mdx* mice (mdx/θ^-/-^) and the deficiency attenuated muscle wasting in the diaphragm [162]. Specifically, there was an increase in myogenin/eMHC positive cells and functional activity as seen by improved running ability [162]. Treating *mdx* mice subject to cardiotoxin injury in the TA and quadriceps femoris with sphingosine-1-phosphate, a T cell inhibitor, increased force production in the EDL. The increase in force is likely due to the reduced fibrosis and fat accumulation seen in the EDL, with a simultaneous increase in myogenic cells and regenerating fibers [163,164].

Lastly, treating *mdx* mice with rapamycin (an immunosuppressant) for six weeks decreased effector CD4^+^/CD8^+^ T cells in the muscle while maintaining Foxp3^+^ T_reg_ cells, which were beneficial to muscle healing [165]. In contrast, one research study did report that depleting CD4^+^ and CD8^+^ T cells at four weeks of age in an *mdx* mouse model had no impact on fibrotic tissue development in the diaphragm, despite a decrease in TGF-β after double CD4^+^/CD8^+^ depletion [166]. The authors suggested that the early time point chosen may have impacted the results as the cells/environment that induce fibrosis could have already been present.

Perforin-expressing CD4^+^/CD8^+^ T cells are one of the main immune cell types found in polymyositis (PM) and dermatomyositis (DM) [134,167,168]. In PM, the inflammatory cells primarily reside in the endomysial space, whereas in DM they reside in the perivascular/perimysial space, but similar levels of CD8^+^ cells are found in both conditions [134]. Both cell types express Fas Ligand in PM, which is associated with high levels of muscle apoptosis [167]. Increased levels of effector T cells, with CD8^+^ T cells being predominant, have also been found infiltrating muscle tissues in patients with juvenile DM syndrome where they localize around blood vessels [168]. In adult DM, while CD8^+^/CD4^+^ T cells are present, it has been suggested that humoral effector mechanisms are the main driver of muscle degeneration, since activated B cells are found in the peripheral blood of patients [134,169].

### 3.5. Role of T_reg_ in Acute and Chronic Muscle Injury

Regulatory T (T_reg_) lymphocytes (Foxp3^+^CD4^+^) have been recognized for their ability to suppress and attenuate inflammation; phenotypically and functionally distinct populations of T_reg_ cells have been identified, such as muscle specific T_reg_ [99,170,171,172,173,174]. Skeletal muscle T_reg_ (mT_reg_) are considered distinct from their lymphoid-organ counterparts by the following criteria: their prevalence, transcriptome, and T cell receptor (TCR) repertoire [99]. mT_reg_ were first documented in 2013 [99] and have a critical role in muscle regeneration following both acute and chronic muscle injuries [99,175]. Expression of Helios and neuropilin-1 (Nrp-1) by mT_reg_ cells showcases that this is a thymic derived T_reg_ cell [99,176,177]. However, mT_reg_ cells have a distinct gene-expression profile, notably upregulating genes encoding T_reg_ mediated suppression (e.g., *IL-10*, *Gzmb, Ctla-4, Tim-3, Klrg1*), tissue repair (e.g., *Il1rl1*, *Areg, Pdgf*), and chemokine receptors (e.g., *Ccr1, Ccr2, Ccr3*), while downregulating genes encoding proteins implicated in the Wnt signaling pathway (e.g., *Tcf7, Lef1, Satb1*), and certain chemokine receptors (e.g., *Cxcr5, Ccr7)* [78,99,170,178,179]. Of the transcripts expressed or repressed by mT_reg_, amphiregulin (Areg), special AT-rich sequence-binding protein-1 (SATB1), and suppression of tumorigenicity 2 (ST2) are of interest due to their role in regenerative activities of the muscle. Areg, a member of the epithelial growth factor family, can directly impact muscle regeneration by promoting MuSC myogenicity and enhancing T_reg_ ability to suppress immune responses [178,180]. SATB1, a chromatin organizer is capable of affecting T_reg_ functionality [99,181,182]. ST2, encoded by Interleukin 1 Receptor Like 1 (Il1rl1), binds interleukin-33 (IL-33) and affects mT_reg_ accumulation and function, promoting effective repair. IL-33 is also known as an alarmin and its levels spike within hours of injury, drive accumulation of T_reg_ in muscle [183]. Ablation of the *Il1rl1* gene is known to prevent IL-33 from interacting with ST2, resulting in impaired T_reg_ recruitment and delayed regeneration [183,184].

Muscle regeneration following an acute muscle injury is influenced by the accumulation of T_reg_ cells at the site of insult. T_reg_ influence macrophage phenotype, modulating muscle regeneration. mT_reg_ gather at the site of injury by day 4, just as the myeloid cell infiltrate switches from a pro- to anti-inflammatory phenotype following acute injuries (e.g., cardiotoxin-induced injury and cryo-injury) [99]. In vitro, macrophages co-cultured with Foxp3^+^CD4^+^CD25^+^ T_reg_ exhibited the ability of T_reg_ to steer monocyte differentiation towards the anti-inflammatory phenotype. T_reg_ produce IL-10, IL-4, and IL-13, cytokines critical in this context for resolving an inflammatory response. Macrophages upregulated the expression of CD206 and CD163, increased the production of CCL18, and enhanced phagocytic capacity when co-cultured with Tregs, these events, with the exception of CD206 upregulation, were partly dependent on IL-10 [98]. It is perhaps paradoxical that IL-4 and IL-13 are critical for an effective T_h_2 response, but the timing, anatomic location and form of stimulus, and concomitant cytokines (namely well-established immunosuppressive IL-10) are important for this division of effector functions. Co-culture of monocytes with T_reg_ also inhibited the ability of monocytes to produce pro-inflammatory cytokines such as TNF-α and IL-6 [185].

In an acute injury model induced by cardiotoxin injection, ablation of T_reg_ resulted in increased production of pro-inflammatory cytokine IFN-γ by NK cells and effector T cells, likely CD8^+^ and T_h_1. The production of IFN-γ led to increased accumulation of the pro-inflammatory macrophage (MHCII^+^) subset and fibrosis. When T_reg_ were present at the injury site, IFN-γ production was limited, promoting the accrual of macrophages and regulating their phenotypic switch [186]. Ablation of T_reg_ in Foxp3-DTR transgenic mice following cardiotoxin injury supported this: the cellular infiltrate was increased, and myeloid cells failed to undergo the phenotypic switch from pro- to anti-inflammatory macrophages. Histologically, T_reg_ ablation led to a disorganized tissue structure with several foci of inflammation, greater collagen deposition, and a decreased number of centrally nucleated fibers [99].

In vitro exposure to activated T_reg_ cells induced MuSC expansion and concomitantly inhibited myogenic differentiation. In vivo studies revealed that the recruitment of T_reg_ cells to acutely injured muscle was limited to the time frame of MuSC expansion [187]. Local expansion of T_reg_ cells is required for skeletal muscle repair, as observed with the results aforementioned, and evident when treatment with an anti-CD25 mAb targeting Foxp3^+^CD4^+^CD25^hi^ T_reg_ cells increased muscle damage in dystrophic mice. Similarly, treatment with complexes of recombinant IL-2 with an anti-IL-2 mAb prevented muscle damage in dystrophic mice while enhancing T_reg_ activity and increasing IL-10 production [99,175,180]. Osteopontin (OPN) is an immunomodulator in *mdx* muscle which promotes fibrosis and expression of TGF-β. Studies ablating OPN concluded that myeloid cells shifted to a pro-regenerative phenotype and led to an increase in intramuscular T_reg_ with reduction in fibrosis [188,189]. Protozoan parasite chronic muscle infections, such as *Toxoplasma gondii*, led to impaired T_reg_-mediated immunomodulation which directly contributes to macrophage-mediated muscle damage. As mentioned above, the restoration of T_reg_ activity rescues muscle regeneration; however, in the case of chronic infection, the suppression of the T_reg_ population promoted regeneration and increased the proportion of M2 macrophages [180,190]. These findings show the complexity of T_reg_ and their function in muscle injury, infection, and disease.

### 3.6. Role of T_reg_ in Aged Muscle

The aging of skeletal muscle is associated with a steady decline in bulk, function, and regenerative capacity due to both intrinsic and environmental factors [191,192]. Poor regeneration is associated with the age-associated decrease in MuSC frequency and function, which can influence T_reg_ recruitment to the muscle [187].

T_reg_ cells increase in lymphoid organs with age; however, they are sparse in aged muscle following injury [183,193]. Following a cardiotoxin-induced injury, T_reg_ accumulation was diminished in the muscle of aged mice, reflecting the poor recruitment, proliferation, and retention of T_reg_ in aged muscle. The exogenous administration of IL-33 restored the T_reg_ population in injured aged muscle, promoting regeneration. IL-33 did not affect the recruitment of T_reg_ to the site of injury; however, it enhanced T_reg_ cell proliferation, these results were observed in both a cardiotoxin-induced injury and cryo-injury [183]. These findings highlight the importance of the IL-33:ST2 axis and its role in mT_reg_ cell function and expansion.

Overall, studies show that a lack of mT_reg_ is partly responsible for the poor muscle regeneration seen in older individuals. Future studies should investigate therapeutic strategies to improve Treg quantity and function in aged muscle.

## 4. Immunomodulators in Skeletal Muscle Regeneration

The inflammatory response should be mitigated under certain conditions to encourage regeneration. Factors that can modulate the immune response have been explored specifically in the context of muscle healing and they include, but are not limited to, mesenchymal stem cell (MSC) therapy with or without a scaffold delivery system and immune suppressant drugs such as FK506 and FTY720 [194].

### 4.1. Immunomodulatory Properties of MSCs

The immunomodulatory properties of MSC have been well documented in application to many different tissues, including but not limited to skeletal muscle, bone, skin, peripheral nerve, blood vessels, cartilage, and tendon [195,196]. Their role in regulating the immune system comes from their ability to secrete a variety of growth factors (e.g., TGF-β1, HGF, prostaglandin 2, SDF-1, NO) and cytokines (e.g., IL-4, IL-6, and IL-10) that suppress pro-inflammatory responses to damage [195]. Specifically, these factors are released upon stem cell activation from pro-inflammatory cytokines and they inhibit T cell proliferation, encourage the shift from a Th_1_ to a Th_2_ response, and impair the maturation of dendritic cells. The shift from a Th_1_ to a Th_2_ response may also cause a shift in macrophage phenotype because the Th_2_ response skews macrophages to a pro-regenerative M2 phenotype [195,196]. They can also impact a significant number of other pro-inflammatory cells including B-cells, and natural killer cells [195]. Collectively, inhibiting these cell types directly impacts the level of inflammation found in the defect area. In a recent clinical study, allogeneic placenta-derived, mesenchymal-like adherent cells were found to increase muscle strength and volume in patients who underwent hip arthroplasty. The authors concluded that the cell therapy was safe and the beneficial effects to injured skeletal muscle were attributed to MSC induced immunomodulation [197].

The impact of MSCs on immune cells has been studied in several ways, including in vitro, in vivo, and in disease models such as muscular dystrophy. When culturing MSCs with CD8^+^ T cells, these cells decreased the expression of CD8, CD28, and CD44, indicating a potential shift in cellular phenotype and function [198]. Additionally, MSCs reduced the T cell release of IFN-γ and granzyme B, which required stimulation of CD8^+^ T cells directly from monocytes, while increasing the release of IL-4 [198,199]. Phenotypic changes were also seen in CD4^+^ T cells that were cultured with bone marrow derived MSCs, and CD4^+^/CD8^+^ T cells downregulated CD25, CD38, and CD69, preventing their activation [200].

In vitro studies involving the co-culture of M0, M1, and M2 macrophages with MSCs demonstrated promising results. Naive macrophages cultured with MSCs were shifted toward an M1 state and had an increased ability to kill pathogens via a boost in the respiratory burst response. In contrast, M1s cultured with MSCs shifted towards an M2 phenotype, as evidenced by increased M2 gene expression. Therefore, MSCs exhibit diverse effects on of macrophages that are largely attributed to a prostaglandin E2 (PGE₂) dependent mechanism [201].

Animals with substantial skeletal muscle damage that were implanted with MSCs showed a significant decrease in pro-inflammatory cytokines (IL-1β, IL-6, TNF-α, TGF-β, and IFN-γ) and an increase in an important anti-inflammatory cytokine IL-10 [202]. The changes seen at the molecular level corresponded to histological differences in muscle regeneration, with less collagen content and fibrosis in MSC treated animals coinciding with improvements in microvasculature formation within the muscle [202]. In another model of VML injury, MSCs were seeded onto an ECM scaffold, which upon implantation led to a shift in macrophage polarization from an M1 to an M2 phenotype, resulting in improved muscle regeneration and reduced fibrotic tissue deposition [203]. In contrast, another study showed that the delivery of MSCs on an ECM scaffold supported functional recovery following VML with little to no muscle fiber regeneration. The authors attributed the improvement in function to a fibrotic scar mediated bridging effect that allowed for increased force transmission vs. production [204].

δ-sarcoglycan null dystrophic hamsters that received an intramuscular injection of MSCs showed a reduction in the upregulation of pro-inflammatory markers in the circulatory system, including immunoglobulin A, vascular cell adhesion molecule-1, and myeloperoxidase [205]. Other beneficial results were seen within the muscle itself, which demonstrated that leukocyte antigens, oxidative stress, and NF-kB levels did not increase in comparison to animals not receiving the MSC treatment [205]. MSC derived nuclei were observed in both center and periphery of myofibers, suggesting their role in fiber growth via donation of nucleus or cell fusion. In another study, MSC administration into dystrophic mice reduced inflammatory cytokines (such as TNF-α, IL-6) and oxidative stress. Simultaneously, the MSC treatment increased VEGF, IL-10, and IL-4 release [206].

Immunomodulatory benefits can also be exhibited by MSC derived exosomes, which are nanovesicles secreted by mammalian cells for intercellular communication [207]. These vesicles contain bioactive molecules such as lipids, proteins, mRNAs and microRNAs (miRNAs) unique to the cell of origin [208,209]. To evaluate the therapeutic potential of MSC-derived exosomes, several clinical trials (http://clinicaltrials.gov, accessed on 10 February 2021) have been completed or are currently ongoing. Exosomes recapitulate the broad therapeutic effects attributed to MSCs [210,211]. For instance, MSC exosomes contain numerous anti-inflammatory and anti-fibrotic miRNAs [211] as well as several cytokines and growth factors including IL-6, IL-10, TGF-β1, VEGF and HGF [212]. These molecules can promote both immunomodulation and tissue regeneration. As an acellular MSC byproduct, exosomes can readily circulate through organs, elicit a minimal immune response, avoid phagocytosis and elicit cellular responses by binding to specific receptors on the target cell [213]. In mice with cardiotoxin injury, MSC exosomes support muscle regeneration, enhanced angiogenesis, and reduced fibrosis [214]. In another study, MSC-derived exosome treatment following cardiotoxin injury increased the expression of markers associated with an M2 macrophage phenotype (Arg1^+^ and Ym1^+^), which coincided with improved muscle regeneration evidenced by increased quantity of myoblasts and fibers with centrally located nuclei [215]. In a murine VML model, co-delivery of muscle ECM and MSC extracellular vesicles (EV) enhanced angiogenesis and myogenesis but reduced fibrosis. A decrease in M1-like markers (iNOS) with an increase in M2-like markers (CD163 and Arg1) was also reported [216]. The same study created a co-culture model with C2C12 cells with macrophages and introduced a cardiotoxin to mimic an injured microenvironment in vitro. Treatment of the in vitro cardiotoxin injury with MSC EVs elicited a protective action against cell apoptosis and increased cell survival as shown by new laminin deposition and increased number of proliferating cells [216].

In conclusion, MSCs display a versatile ability to enhance the pro-regenerative environment by reducing inflammation and encouraging myofiber formation. The molecular impact of MSCs on the cellular niche of regenerating muscle has also improved functional deficits seen with muscle damage.

### 4.2. Immunosuppressants as Therapies for Muscle Regeneration

Immunosuppressant drugs, such as FK506 and FTY720, have been used for modulation of the immune response in skeletal muscle. FTY720 is a sphingosine-1-phosphate receptor modulator that can regulate chronic inflammation and fibrosis deposition [217]. FK506, also known as tacrolimus, is a macrolide antibiotic that acts on T cells and can suppress their immune response [218]. In a model of limb-girdle type 2C muscular dystrophy, FTY720 was administered for 3 weeks to mice aged 3 weeks and greatly reduced muscle membrane permeability and fibrosis [219]. Additionally, it also increased the upregulation of sarcoglycan which could lead to the protection of the sarcolemma from shear forces. Therefore, the drug could partly protect the muscle from the damaging effects of inflammation commonly associated with the disease. In another study conducted on FTY720, mice were subjected to hind limb ischemia-reperfusion muscle injuries and then subsequently treated with the drug [220]. Although local effects of muscle regeneration (i.e., cross sectional area of muscles, myofibers with CLN, etc.) were not measured, the study revealed that FTY720 will reduce systemic inflammation by causing a decrease in peripheral blood T cell levels. In conjunction with the T cell decrease, a significant reduction in serum creatinine and important cytokines, such as TNF-α, IL-6, IL-10, and IL-18, were also observed.

In complex models of VML that include bone fractures, or osteotomies, the use of FK506 induced better healing in bone fractures that correlated with changes in cell types in the overlying, injured musculature [10]. Bone defects compounded by VML of the TA are difficult to treat. Introducing the FK506 drug reduced the numbers of T-lymphocytes and macrophages found in the TA, with reduced T-lymphocytes in the callus of the bone [10]. Administering the drug did not improve TA force production, but it did restore mechanical properties of the bone in the presence of the muscle defect [10]. These findings suggest that the drug’s impact on the injured muscle, such as the reduction of T cell infiltration into all areas of damage, strongly and positively impact the healing ability of the bone. Regulating the immune response with FK506 delivered in conjunction with a MMG to a VML injury in swine muscle led to a marginal improvement of force deficit with a higher fraction of muscle fibers vs. fibrotic tissue [221].

## 5. Conclusions

The physiological repair mechanisms following skeletal muscle injury or disease are currently under investigation. Cells of the innate and adaptive immune system play diverse and complex roles in muscle healing, and the interplay between endogenous stem cell populations and infiltrating immune cells likely determines the regenerative outcome. Greater insight into these cellular interactions will pave the way for new therapeutic strategies. Recent substantial progress in just identifying the cells present in healthy muscle and regenerating skeletal muscle should help directing impactful hypotheses in this area [222,223,224,225]. Mounting evidence indicates that immune cell subsets can have both pro- and anti-reparative functions. The environmental cues and molecular triggers which control the switch between pro- and anti-reparative functions need further investigation. Some studies suggest that a transient and controlled immune response is critical to muscle healing, but an overactive and persistent immune response can be detrimental to tissue repair. Since the clinical treatment options for muscle injuries are scarce, modulation of the immune response might present an effective approach to boost the innate regenerative capacity of the skeletal muscle niche.

## Figures and Tables

**Figure 1 ijms-22-03265-f001:**
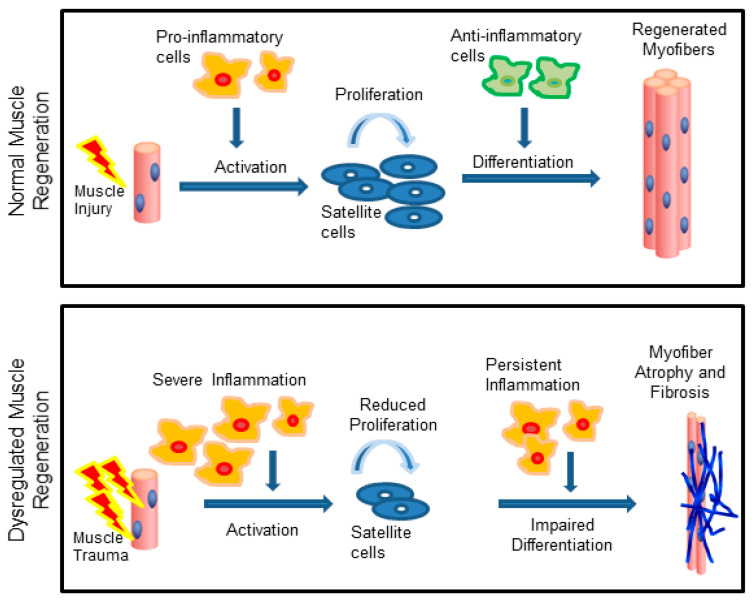
Skeletal muscle regeneration is dependent on the inflammatory response. Following acute injuries, the pro-inflammatory cells support satellite cell proliferation, while anti-inflammatory cells support differentiation (**top**). In chronic injuries, persistent inflammation impairs satellite cell activity resulting in muscle wasting and fibrosis (**bottom**). Adapted from [7].

**Figure 2 ijms-22-03265-f002:**
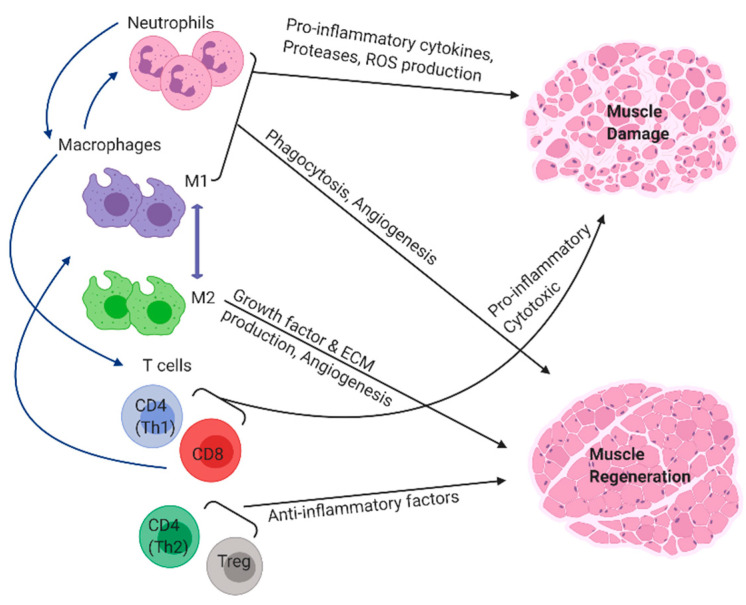
Inflammatory cells of the innate and adaptive immune system participate in the process of muscle regeneration and repair. Blue arrows indicate immune cells recruiting each other through the secretion of various soluble mediators.

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
