# Peer review of "The Role of Innate and Adaptive Immune Cells in Skeletal Muscle Regeneration"

_ijms, 2021, doi:10.3390/ijms22063265_

Round 1

Reviewer 1 Report

The authors have provided a thorough and insightful overview on the complexity of the innate and adaptive immune system on muscle regeneration and highlighted the importance of the cellular interactions among the immune cells or with the MSCs in muscle healing. Further studies looking into the molecular switches between the anti-inflammatory and pro-reparative phenotypes are required for developing new therapeutics for treating muscle damage.

Specific comments:

  1. For Figure 2, there seem to be 2 shades of the blue arrows. Could this be further clarified or modified?
  2. Please proofread on minor spelling mistakes. 

Author Response

We thank the reviewer for taking the time to review our manuscript. The reviews were helpful and have improved the quality of our work. Please see our response to the reviewer comments below:

Specific comments:

  1. For Figure 2, there seem to be 2 shades of the blue arrows. Could this be further clarified or modified?

Author Response: We thank the reviewer for the comment. We have edited Figure 2. The shade of blue arrows is now consistent. We have also edited the figure legend to clarify what the blue arrows indicate.

  1. Please proofread on minor spelling mistakes. 

Author Response: We have edited the manuscript.

Reviewer 2 Report

In the current manuscript, Ziemkiewicz et al. reviewed the roles of immune cells in skeletal muscle regeneration, with a focus on neutrophils, macrophages, and T cells. The authors emphasized how pro-inflammatory and anti-inflammatory factors (M1 vs M2 macrophages, cytotoxic/helper T cells vs. Tregs) are precisely regulated during the process of injury recovery or in the context of aging. The review is very comprehensive and the manuscript is well written. I only have a few minor suggestions.

Figure 2, blue lines/arrows are not showing properly and can be confusing. Do they mean interactions or recruitment? Please add notes in the figure or legends. Additionally, figure 2 can be moved to the end as a summary after all the cell types have been discussed. 

Line 41. References should be added for the recruitment of T cells by macrophages during muscle regeneration.

Line 328. First appearance of VML, full term (or even a brief introduction) is needed.

Line 387. The word "any" should be removed

For T helper cells, since different subgroups (Th1, Th2, Th17 ect.) have distinct roles, it may be helpful to have a brief description of these cell types.

Author Response

We thank the reviewer for taking the time to review our manuscript. The reviews were helpful and have improved the quality of our work. Please see our response to the reviewer comments below:

  • Figure 2, blue lines/arrows are not showing properly and can be confusing. Do they mean interactions or recruitment? Please add notes in the figure or legends. Additionally, figure 2 can be moved to the end as a summary after all the cell types have been discussed. 

Author Response: We have edited Figure 2. We have also edited the figure legend to clarify that the blue arrows indicate recruitment.

  • Line 41. References should be added for the recruitment of T cells by macrophages during muscle regeneration.

Author Response: We thank the reviewer for the comment. We have added a relevant reference to the second paragraph of the introduction.

Reviewer 3 Report

This article has been written well and covered most of the current knowledge that we know. If it is accepted, the information would be useful to the audience.

Author Response

Comments and Suggestions for Authors

This article has been written well and covered most of the current knowledge that we know. If it is accepted, the information would be useful to the audience.

Author Response: We thank the reviewer for the comment.